# Histone deacetylase HDA-1 modulates mitochondrial stress response and longevity

Li-Wa Shao[1,2,4], Qi Peng [3,4], Mingyue Dong[1,2], Kaiyu Gao[1], Yumei Li [3], Yi Li[1,2], Chuan-Yun Li [3✉] & Ying Liu [1✉]

The ability to detect, respond and adapt to mitochondrial stress ensures the development and survival of organisms. *Caenorhabditis elegans* responds to mitochondrial stress by activating the mitochondrial unfolded protein response (UPR$^{mt}$) to buffer the mitochondrial folding environment, rewire the metabolic state, and promote innate immunity and lifespan extension. Here we show that HDA-1, the *C. elegans* ortholog of mammalian histone deacetylase (HDAC) is required for mitochondrial stress-mediated activation of UPR$^{mt}$. HDA-1 interacts and coordinates with the genome organizer DVE-1 to induce the transcription of a broad spectrum of UPR$^{mt}$, innate immune response and metabolic reprogramming genes. In rhesus monkey and human tissues, HDAC1/2 transcript levels correlate with the expression of UPR$^{mt}$ genes. Knocking down or pharmacological inhibition of HDAC1/2 disrupts the activation of the UPR$^{mt}$ and the mitochondrial network in mammalian cells. Our results underscore an evolutionarily conserved mechanism of HDAC1/2 in modulating mitochondrial homeostasis and regulating longevity.

[1] Beijing Advanced Innovation Center for Genomics, State Key Laboratory of Membrane Biology, Beijing Key Laboratory of Cardiometabolic Molecular Medicine, Institute of Molecular Medicine, Peking University, 100871 Beijing, China. [2] Peking-Tsinghua Center for Life Sciences, Academy for Advanced Interdisciplinary Studies, Peking University, 100871 Beijing, China. [3] Laboratory of Bioinformatics and Genomic Medicine, Beijing Key Laboratory of Cardiometabolic Molecular Medicine, Institute of Molecular Medicine, Peking University, 100871 Beijing, China. [4]These authors contributed equally: Li-Wa Shao, Qi Peng. ✉email: chuanyunli@pku.edu.cn; ying.liu@pku.edu.cn

Mitochondria play essential and pervasive roles in biology. Originating from endosymbiosis of proteobacteria, these organelles not only provide host cells with ATP generated through oxidative phosphorylation, but also participate in numerous biological processes from the regulation of calcium homeostasis to innate immune responses and programmed cell death[1]. The normal function of mitochondria is challenged by intrinsic stimuli and by extrinsic pathogens and xenobiotics[2]. Mitochondrial dysfunction has been extensively linked with aging[3,4] and numerous diseases such as neurodegenerative disorders[5]. Given the importance of mitochondria, cells employ quality control mechanisms to actively surveil mitochondrial function and initiate protective programs upon mitochondrial damage.

One such quality control mechanism is the mitochondrial unfolded protein response (UPR$^{mt}$), which relays mitochondrial stress signals to the transcription of nuclear-encoded genes that protect mitochondria, such as chaperones and proteases[6–9]. In *C. elegans*, the transcriptional response is mainly governed by two transcription factors, ATFS-1 and DVE-1[7,9–11]. In addition, growing evidence has also suggested that chromatin modifiers act in conjunction with these two transcription factors to modulate UPR$^{mt}$ activation[12–14].

Eukaryotic chromosomes are packed into three-dimensional higher-order structures. Chromatin modifier proteins induce changes in chromatin architecture and thereby control accessibility of DNA sequences to the transcription machinery. For example, post-translational modifications, such as acetylation, of histones H3 and H4 have been reported to dictate the active or inactive chromatin state and ultimately affect gene expression[15,16]. Histone acetylation states are tightly regulated by two chromatin modifiers with opposing function, histone acetyltransferase (HAT) and histone deacetylase (HDAC)[16], allowing for the precise control of gene expression. Given that UPR$^{mt}$ requires the activation of a broad spectrum of genes to counteract mitochondrial stress and reset the cellular metabolic state, the biological importance of chromatin modifier proteins in UPR$^{mt}$ warrants further exploration.

Here, we report that histone deacetylase HDA-1 is a key regulator of UPR$^{mt}$ in *C. elegans*. Mechanistically, HDA-1 acts in concert with DVE-1 to promote the transcription of genes involved in the mitochondrial stress response, innate immune response, and metabolism. Moreover, expression profiles in tissues from rhesus monkey and human indicate that transcript levels of primate HDAC1/2 strongly correlate with the expression of UPR$^{mt}$ genes. Collectively, our results highlight the conserved and crucial function of HDAC1/2 in regulating the mitochondrial stress response and its beneficial outcomes.

## Results

**hda-1 is required for UPR$^{mt}$ activation.** To understand the molecular mechanisms that govern activation of UPR$^{mt}$, we carried out a genome-wide RNA interference (RNAi) screen by employing *C. elegans* UPR$^{mt}$ reporter strains such as *hsp-6p::gfp*[2]. Specifically, we fed age-synchronized wild-type worms expressing GFP fluorescent reporters with dsRNA-expressing bacteria and tested the animals for their abilities to activate GFP expression upon mitochondrial perturbation. *hda-1*, one of the HDAC genes, was recovered from this screen[2]. RNAi-mediated knockdown of *hda-1* impaired the activation of the UPR$^{mt}$ reporter *hsp-6p::gfp* that is induced by RNAi of the nuclear-encoded mitochondrial gene *atp-2* (ATP synthase F1 subunit beta) or *cco-1* (cytochrome c oxidase subunit 5B) (Fig. 1a, b). Deficiency of *hda-1* also suppressed the elevation of endogenous transcript levels of *hsp-6* under mitochondrial stress (Fig. 1c). In addition, lack of *hda-1*

impaired the activation of another UPR$^{mt}$ reporter, *hsp-60p::gfp*, during mitochondrial perturbation (Fig. 1d). To test if *hda-1* is specifically required for UPR$^{mt}$ activation, or if it actually mediates a more general stress response, we knocked down *hda-1* in the endoplasmic reticulum (ER) stress reporter strain *hsp-4p::gfp* or the heat shock stress reporter strain *hsp-16.2p::gfp*, and challenged the animals with either the ER inhibitor tunicamycin or heat shock treatment. Deficiency of *hda-1* did not affect the activation of ER stress or heat shock stress response (Fig. 1e, f; Supplementary Fig. 1a, b). Collectively, these results indicate that *hda-1* plays a specific role in modulating the mitochondrial stress response.

HDAC plays an evolutionarily conserved role in removing acetyl moieties from core histones. Based on their structures and functions, HDACs have been grouped into three classes: class I and II HDACs share sequence homology in their catalytic domain[17,18], while class III HDACs, as exemplified by SIR2 (silent information regulator 2), have a unique feature and require NAD$^+$ as a cofactor for catalysis[19–23]. *C. elegans* class I HDACs include *hda-1*, *hda-2*, and *hda-3*, and class II HDACs include *hda-4*, *hda-5*, *hda-6*, *hda-10*, and *hda-11*[24,25]. Considering the high homology between class I and II HDACs, we sought to test if any other HDAC in *C. elegans* plays a similar role to *hda-1* in mediating the mitochondrial stress response. We knocked down individual class I and II HDACs in *C. elegans*, and tested the animals' ability to induce UPR$^{mt}$ upon mitochondrial perturbation. Interestingly, knockdown of other HDACs failed to suppress *atp-2* RNAi-induced UPR$^{mt}$ (Supplementary Fig. 1c, d), suggesting a specific role of *hda-1* in mediating UPR$^{mt}$.

Next, we generated transgenic strains expressing HDA-1::GFP fusion protein driven by the *hda-1* promoter. Notably, we found that *hda-1* expression was elevated in the nuclei of *C. elegans* intestinal cells under *cco-1* or atp-2 RNAi treatment (Supplementary Fig. 1e, f). In addition, overexpression of HDA-1 slightly upregulated the basal level of UPR$^{mt}$ and further elevated the induction of UPR$^{mt}$ under mitochondrial stress conditions (Supplementary Fig. 1g, h).

To further validate the function of *hda-1* in UPR$^{mt}$, we collected total RNAs from wild-type animals or *hda-1*-deficient animals in the presence or absence of mitochondrial perturbation, and performed RNA sequencing (RNA-seq) analysis to characterize *hda-1*-dependent genes during the mitochondrial stress response. 805 genes were significantly upregulated in wild-type animals under *atp-2* RNAi treatment. Among them, 283 genes were significantly less induced in *hda-1*-deficient animals (Fig. 1g; Supplementary Data 1). Gene ontology (GO) functional enrichment analysis revealed that these *hda-1*-dependent genes are enriched in GO Biological Processes such as Metabolic process, Response to stress, Immune response and Cellular detoxification (Fig. 1h and Supplementary Data 2). This further supports a crucial role of *hda-1* in modulating the mitochondrial stress response.

**HDA-1 interacts with DVE-1 to regulate UPRmt.** To explore the molecular mechanism by which HDA-1 regulates the mitochondrial stress response, we immunoprecipitated HDA-1 from *C. elegans* expressing HDA-1::GFP using anti-GFP antibody and performed mass spectrometry analysis to search for HDA-1-interacting proteins. Notably, we found that HDA-1 interacts with the homeodomain-containing transcription factor DVE-1, a well-known component of the *C. elegans* UPR$^{mt}$ pathway[7], under control RNAi or *atp-2* RNAi treatment that perturbs mitochondrial function. Immunoprecipitation experiments on lysates from a transgenic *C. elegans* strain expressing both FLAG-tagged HDA-1 and GFP-tagged DVE-1 further validated the interaction between these two proteins (Fig. 2a).

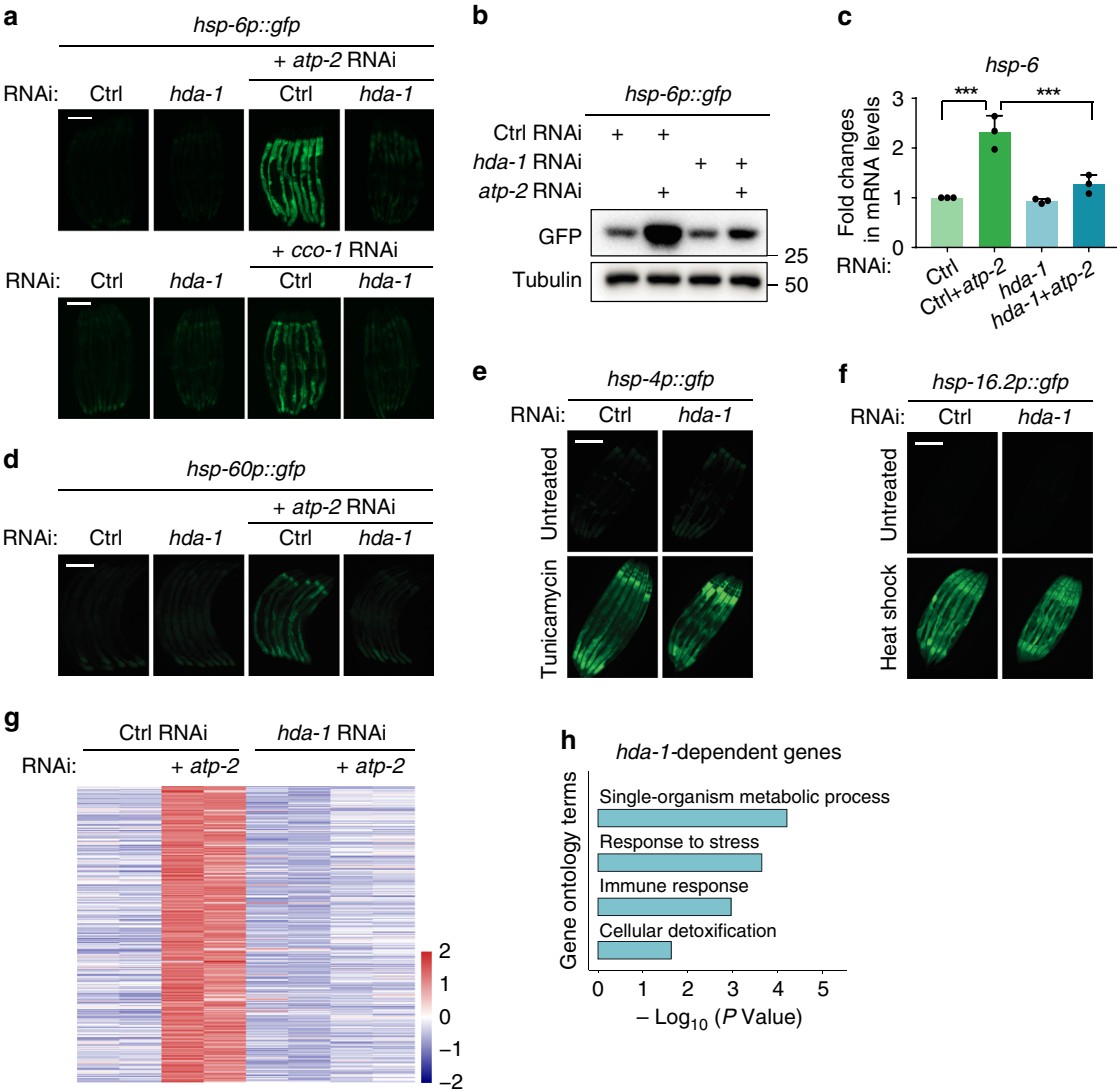

**Fig. 1 *hda-1* is required for UPR<sup>mt</sup> activation. a** Representative fluorescence images of *hsp-6p::gfp* worms. Worms on control (ctrl) or *hda-1* RNAi were untreated or treated with *atp-2* RNAi or *cco-1* RNAi. Scale bar, 200 μm. **b** Immunoblotting of GFP levels in *hsp-6p::gfp* worms. Tubulin serves as a loading control. **c** qRT-PCR measures endogenous transcript levels of *hsp-6* in worms ($n = 3$ independent experiments, $n ≈ 1000$ worms per sample). **d** Representative fluorescence images of *hsp-60p::gfp* worms. Scale bar, 200 μm. **e** Representative fluorescence images of *hsp-4p::gfp* worms. Worms were fed with control or *hda-1* RNAi and untreated or treated with tunicamycin. Scale bar, 200 μm. **f** Representative fluorescence images of *hsp-16.2p::gfp* worms. Worms on control or *hda-1* RNAi were untreated or treated with heat shock. Scale bar, 200 μm. **g** Heat map showing the expression pattern of 283 genes whose expression was upregulated under mitochondrial stress. Genes with an adjusted p value < 0.05 calculated by Cuffdiff were selected as differentially expressed genes. The heat map is scaled by row and colored according to the z-score. Genes with a higher expression level than the mean are colored red; genes with a lower expression level than the mean are colored blue. **h** Gene ontology (GO) term analysis of the 283 *hda-1*-dependent genes, whose expression levels were upregulated during UPR<sup>mt</sup>. Results in (**c**) are shown as mean + SD, and *P* values were calculated by one-way ANOVA and Tukey's multiple comparisons test (***$P < 0.001$). In (**h**), *P* values are DAVID modified Fisher exact *P* values, one-sided. Source data are provided as a Source Data file.

To further characterize the relationship between HDA-1 and DVE-1, we examined the expression profile of HDA-1 in *hda-1p::hda-1::gfp* animals. Interestingly, we noticed that the protein level of HDA-1 was dramatically reduced by *dve-1* RNAi (Fig. 2b and Supplementary Fig. 2a). The reduction of HDA-1 protein level under *dve-1*-deficient conditions seems to be mediated by ubiquitin-mediated degradation, because RNAi knockdown of both *ubq-1* and *ubq-2*, the only two ubiquitin-encoding genes in *C. elegans*[26], abolished the reduction of HDA-1 upon *dve-1* RNAi treatment (Supplementary Fig. 2b). Conversely, knockdown of HDA-1 also reduced the DVE-1 protein level (Supplementary Fig. 2c), while overexpression of HDA-1 elevated the protein level of DVE-1 (Fig. 2c). Furthermore, qPCR analysis of *hda-1* or *dve-1*

transcript levels in the presence or absence of the other partner revealed that the transcription of *hda-1* or *dve-1* is not affected by the loss of its partner protein (Supplementary Fig. 2d). Taken together, these results suggest that HDA-1 and DVE-1 may interact and stabilize each other.

Like the human protein SATB1, with which it shares high homology[27], DVE-1 has been reported to show a 'cage-like' distribution that surrounds heterochromatin[12]. Consistent with the interaction between HDA-1 and DVE-1, HDA-1 showed a similar nuclear distribution pattern in the intestinal cells of *C. elegans*, and formed dense structures that colocalized with DVE-1 to surround chromatin regions stained with DAPI (Fig. 2d, e). To further investigate if HDA-1 and DVE-1 regulate the same group

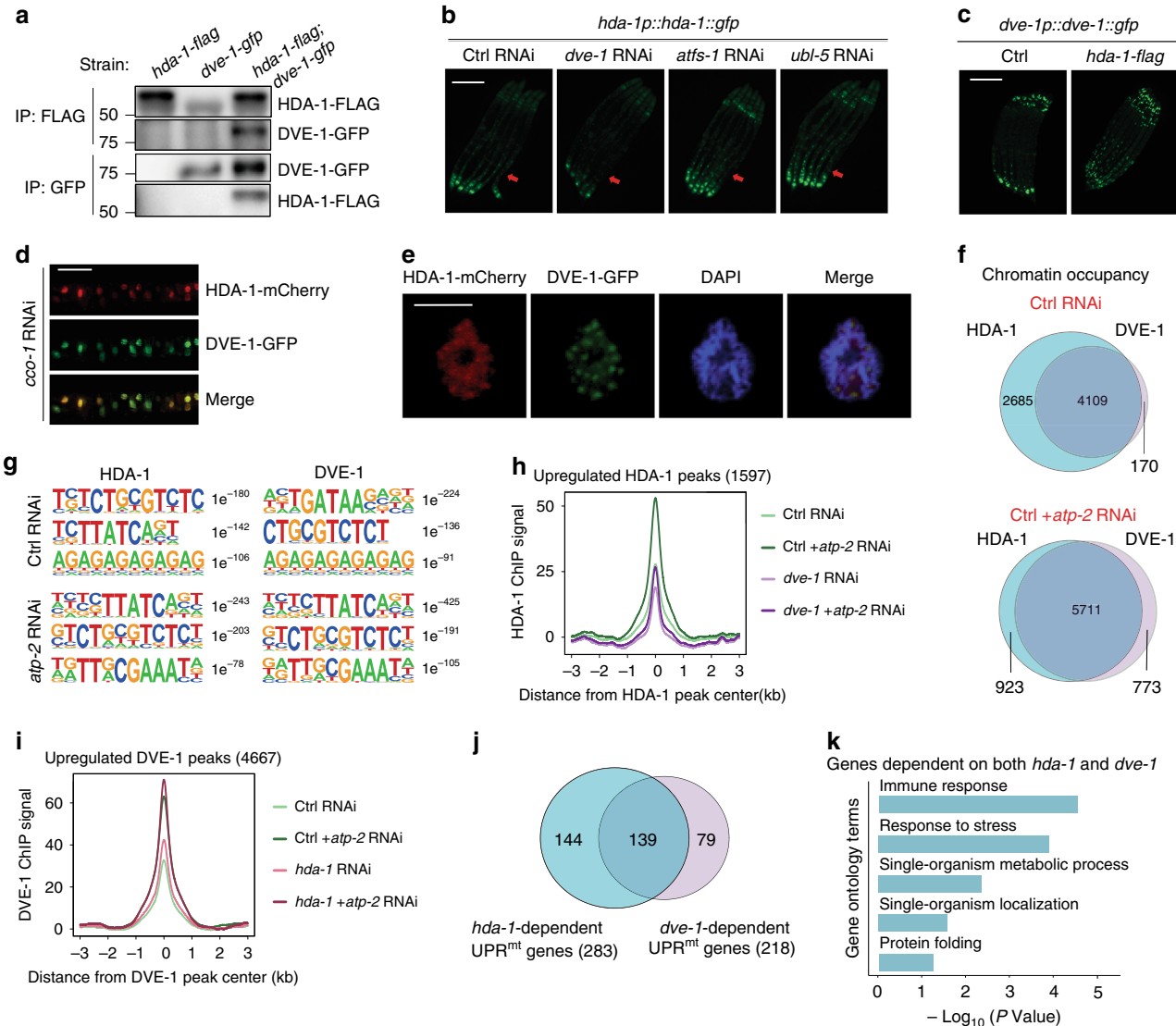

**Fig. 2 HDA-1 interacts with DVE-1 to regulate UPR^mt. a** Immunoprecipitation followed by immunoblotting reveals that HDA-1 interacts with DVE-1 in worms. **b** Representative fluorescence images of *hda-1p::hda-1::gfp* worms. Worms were fed with control, *dve-1*, *atfs-1*, or *ubl-5* RNAi. Images were taken when animals reached young adult stage. Red arrows indicate the posterior region of the intestine where *hda-1p::hda-1::gfp* is induced or suppressed. Scale bar, 200 μm. **c** Representative fluorescence images of *dve-1p::dve-1::gfp* worms, without (control) or with over-expression of *hda-1p::hda-1::flag*. Scale bar, 200 μm. **d**, **e** Representative fluorescence images of *hda-1p::hda-1::mCherry; dve-1p::dve-1::gfp* worms. Worms were fed with *cco-1* RNAi from L1 stage and imaged on day 1 of adulthood. **d** Fluorescence images show the bottom part of the intestine. Scale bar, 50 μm. **e** Fluorescence images reveal the nucleus of an intestinal cell. Scale bar, 5 μm. **f** Venn diagram showing overlap between HDA-1 (blue) and DVE-1 (purple) ChIP peaks in worms fed on control or *atp-2* RNAi. **g** Nucleotide binding motifs of HDA-1 and DVE-1 in the presence or absence of mitochondrial stress. The P values of the motifs are also shown. **h** Analysis of HDA-1 ChIP peak signals in control worms or *dve-1* RNAi worms in the presence or absence of *atp-2* RNAi-induced mitochondrial stress. **i** Analysis of DVE-1 ChIP peak signals in control worms or *hda-1* RNAi worms in the presence or absence of *atp-2* RNAi-induced mitochondrial stress. **j** Venn diagram comparing genes upregulated in response to mitochondrial perturbation that are dependent on *hda-1* (blue) or *dve-1* (purple). **k** GO term analysis of UPR^mt-upregulated genes that are dependent on both *hda-1* and *dve-1*. P values are DAVID modified Fisher exact P values, one-sided. Source data are provided as a Source Data file.

of genes, we carried out chromatin immunoprecipitation followed by sequencing (ChIP-seq) to identify candidate genes regulated by HDA-1- or DVE-1 in the presence or absence of mitochondrial stress. 6794 HDA-1-enriched peaks were uncovered by ChIP-seq analysis under normal conditions. Among these peaks, 60.5% (4,109 out of 6,794) were also recovered in DVE-1 ChIP-seq (Fig. 2f), an overlap significantly higher than the expectation (*Monte Carlo P* value < 0.0001). More importantly, after mitochondrial perturbation, the overlap between HDA-1 and DVE-1 occupancies significantly increased to 86.1% (5711 out of 6634) (Fig. 2f). Intrigued by the large overlap between HDA-1 and

DVE-1 binding loci, we determined the consensus binding motifs of HDA-1 and DVE-1 in the presence or absence of mitochondrial stress (Fig. 2g). Interestingly, HDA-1 and DVE-1 shared more binding motifs upon mitochondrial perturbation, which is consistent with the increased overlap of binding peaks under stress condition (Fig. 2f, g). RNA-seq analysis revealed that 284 and 374 genes were coregulated by HDA-1 and DVE-1 under normal conditions or mitochondrial stress (Supplementary Fig. 2e). We further analyzed the ChIP-seq results and found that the binding of HDA-1 to its target loci may depend on DVE-1, as knockdown DVE-1 suppresses the upregulation of HDA-1

peaks (Fig. 2h and Supplementary Data 3). Conversely, knockdown of HDA-1 did not affect the upregulation of DVE-1 peaks (Fig. 2i and Supplementary Data 3). Consistent with this result, it has been reported that SATB1, the DVE-1 homolog in higher eukaryotes, provides a docking site to recruit the histone deacetylase HDAC1 onto SATB1 target sequences[28].

Further analysis of the RNA-seq results indicated that 283 and 218 genes induced under mitochondrial stress conditions in *C. elegans* were dependent on *hda-1* and *dve-1*, respectively (Fig. 2j). Among the 283 stress-activated *hda-1*-dependent genes, 169 genes (59.7%) contain HDA-1 ChIP peaks within ±500 bp of the transcription start site (Supplementary Data 1). In addition, among the 218 stress-activated *dve-1*-dependent genes, 175 genes (80.3%) contain DVE-1 ChIP peaks within ±500 bp of the transcription start site (Supplementary Data 1). More importantly, 139 genes required both *hda-1* and *dve-1* for induction, accounting for 49.1% of *hda-1*-dependent genes and 63.8% of *dve-1*-dependent genes. These 139 genes that depend on both *hda-1* and *dve-1* were enriched for the GO terms Immune response, Response to stress, Metabolic process and Protein folding (Fig. 2k). We then employed qPCR to validate some of the key genes involved in HDA-1- and DVE-1-dependent regulation. We selected genes associated with some of the most strongly enriched GO terms (Fig. 2k). Knockdown of *hda-1* or *dve-1* significantly suppressed the induction of *cdr-4*, *pgp-1*, *nhr-115*, *M04C3.2* (GO term: Immune response), *dnj-10*, *djr-1.2* (GO term: Stress response), *cyp-33C8*, *ipla-3*, *hmgs-1*, *tars-1* (GO term: Single-organism metabolic process) and *cua-1* (GO term: Single-organism localization) upon mitochondrial stress (Supplementary Fig. 3a). Moreover, a hypomorphic allele of *hda-1* also suppressed the induction of stress response genes (Supplementary Fig. 3b). Taken together, these results suggested a role of HDA-1, in coordination with DVE-1, to activate UPR$^{mt}$ and UPR$^{mt}$-induced innate immune response and metabolic reprogramming.

**HDA-1 is required for UPRmt-mediated innate immunity.** Mitochondrial function is greatly challenged by pathogens and xenobiotics that metazoans encounter in their natural habitats[2]. As a surveillance mechanism that defends against natural infection, UPR$^{mt}$ not only initiates mitochondrial protective responses to induce the expression of mitochondrial chaperones and proteases, but also activates innate immune responses. Our finding that HDA-1 regulates genes related to the GO terms Immune response and Metabolic process led us to further examine its critical function in animal fitness during mitochondrial stress. We first employed *irg-1p::gfp* transgenic animals, a reporter strain for pathogen-infected response and an indicator for the induction of innate immune response[29]. The *irg-1p::gfp* reporter worms were challenged with a *Pseudomonas aeruginosa* strain isolated from natural habitats harboring wild *C. elegans* populations, which has been shown to perturb mitochondrial function and induce expression of the UPR$^{mt}$ reporter *hsp-6p::gfp*[2]. In wild-type animals, *irg-1p::gfp* expression was induced upon *Pseudomonas* infection, whereas deficiency of *hda-1* significantly suppressed the induction of *irg-1*, to a similar extent as deficiency of *dve-1* (Fig. 3a; Supplementary Fig. 4a, b). Knockdown of *hda-1* or *dve-1* also suppressed the induction of several other immune response genes upon *P. aeruginosa* infection (Fig. 3b). In addition, deficiency of *hda-1* or *dve-1* reduced the survival rate of worms and promoted the accumulation of *P. aeruginosa* in *C. elegans* when they were exposed to *Pseudomonas* (Fig. 3c–e and Supplementary Fig. 4c, d)[30]. Conversely, overexpression of HDA-1 or DVE-1 promoted animal survival and reduced the accumulation of *P. aeruginosa* upon pathogen infection (Fig. 3f and Supplementary Fig. 4 f, e).

Next, we asked if the function of HDA-1 can be partially mediated by its target genes. We tested whether *hmgs-1*, one of the 139 genes coregulated by HDA-1 and DVE-1, can modulate the innate immune response upon *P. aeruginosa* infection. *hmgs-1* encodes an HMG-CoA synthase, which has been shown to be required for the induction of UPR$^{mt}$ [2]. Knocking down *hmgs-1* by RNAi reduced the survival rate of worms and promoted the accumulation of *P. aeruginosa* when worms were exposed to this pathogen (Supplementary Fig. 4g, h), similar to the effect observed under *hda-1* RNAi. However, overexpression of *hmgs-1* could not rescue the loss of immunity caused by *hda-1* knockdown (Supplementary Fig. 4i, j), which suggests that expression of this target gene alone is not sufficient to rescue phenotypes caused by *hda-1* deficiency.

Moreover, *hda-1* or *dve-1* RNAi also suppressed the activation of the immune response and reduced the survival rate when the animals were challenged with another mitochondrial insult, a *Rhodococcus* strain isolated from the natural habitat of *C. elegans* (Fig. 3g, h; Supplementary Fig. 4k). Collectively, these results indicate that *hda-1* is required for UPR$^{mt}$-activated innate immune response.

**HDA-1 affects animal aging and age-related pathology.** Genes important for mitochondrial function have been recovered from a genome-wide RNAi screen in *C. elegans* for determining worm lifespans[4]. Moreover, perturbations of mitochondrial electron transport in various species have been shown to extend their lifespans[3,31–33]. Since HDA-1 plays a critical role in mitochondrial stress response, we suspected that it may also play a role in mitochondrial stress-induced lifespan extension. Therefore, we examined the lifespans of wild-type or *hda-1*-deficient worms in the presence or absence of mitochondrial perturbation. Interestingly, *hda-1* RNAi greatly reduced the lifespan extension induced by feeding worms with *atp-2* RNAi to inhibit mitochondrial function (Fig. 4a). We noticed that *hda-1* RNAi, in the absence of mitochondrial perturbation, shortened worm lifespan (Fig. 4a). However, a previous study reported that *hda-1* RNAi does not affect *C. elegans* lifespan[34]. We speculated that there may be two reasons for this discrepancy: (1) different IPTG induction times were used for dsRNA expression, which may affect *hda-1* RNAi efficiency (Supplementary Fig. 5a); (2) the previous study used liquid culture to carry out lifespan analysis, whereas we used solid agar. To further dissect the function of *hda-1* in lifespan regulation, we measured the lifespan of worms carrying the *hda-1* (*e1795*) hypomorphic allele and found that this hypomorphic mutation also shortened worm lifespan (Supplementary Fig. 5b). It should be noted that we also observed a shortened lifespan when *eat-2*(*ad1116*) mutant animals were fed with *hda-1* RNAi (Supplementary Fig. 5c). *eat-2*(*ad1116*) mutants had reduced pharyngeal pumping, therefore mimicking the caloric restriction effect and promoting lifespan extension[35]. It has been reported that *eat-2* animals have decreased mitochondrial potentials[36] and activate the ZIP-2 pathway, which contributes to the improvement of mitochondrial integrity[37]. Therefore, it will be interesting to understand if *hda-1* regulates lifespan through maintenance of mitochondrial integrity, or through a more general mechanism.

A progressive decline of responses toward proteostatic stress occurs with aging, leading to the toxic accumulation of protein aggregates, such as proteins containing polyglutamine (polyQ) repeats[38]. In line with the role of HDA-1 in mediating aging responses, knockdown of *hda-1* enhanced polyQ toxicity and greatly impaired animal movement (Fig. 4b). Conversely, overexpression of HDA-1 suppressed polyQ toxicity and improved animal movement (Fig. 4c). In addition, knockdown of *hda-1* increased the number of polyQ aggregates in *C. elegans* muscle

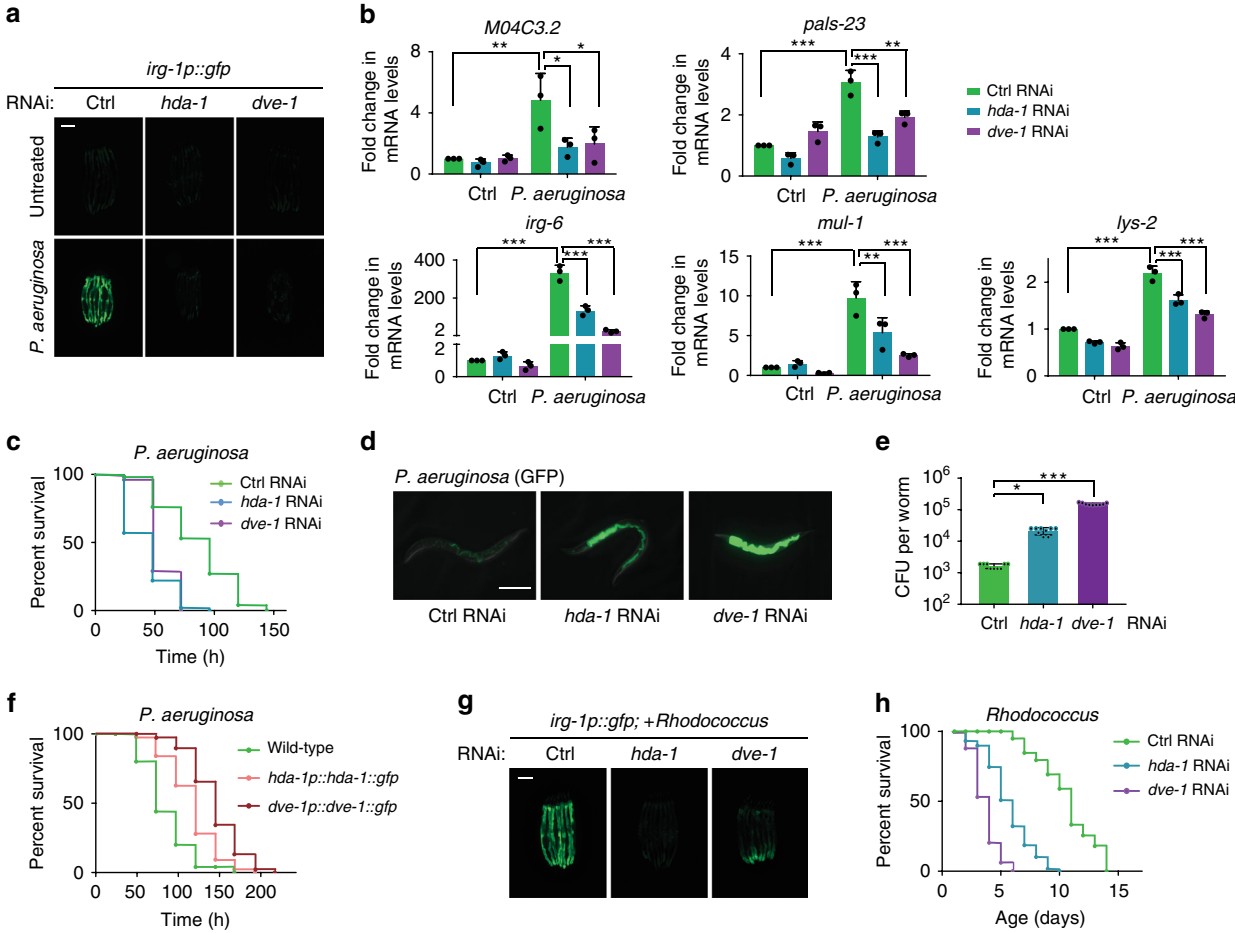

**Fig. 3 HDA-1 is required for UPR<sup>mt</sup>-mediated innate immunity. a** Representative fluorescence images of *irg-1p::gfp* worms. Worms on control, *hda-1* or *dve-1* RNAi were untreated or exposed to *P. aeruginosa*. Scale bar, 200 μm. **b** qRT-PCR measurement of the endogenous mRNA levels of immune response genes in wild-type worms raised on control RNAi, *hda-1* RNAi or *dve-1* RNAi and treated with *P. aeruginosa*. (*n* = 3 independent experiments, *n* ≈ 1000 worms per sample). **c** Survival curves of control, *hda-1*, or *dve-1* RNAi worms in the *P. aeruginosa* slow-killing assay. *n* = 50 worms for each condition. **d** Representative fluorescence images showing accumulation of *P. aeruginosa* (GFP) in intestines of worms fed on control, *hda-1* or *dve-1* RNAi. Scale bar, 200 μm. **e** CFU (colony forming units) were quantified for experiments in (**d**). *n* = 30 worms for each sample. **f** Survival curves of wild-type, *hda-1p::hda-1::gfp*, or *dve-1p::dve-1::gfp* worms in the *P. aeruginosa* slow-killing assay. *n* = 50 worms for each sample. **g** Representative fluorescence images of *irg-1p::gfp* worms raised on control, *hda-1* or *dve-1* RNAi and treated with *Rhodococcus*. Scale bar, 200 μm. **h** Survival curves of control, *hda-1*, or *dve-1* RNAi worms in the *Rhodococcus* killing assay. *n* = 55, 60, 59 worms respectively for each sample. Results in (**b**) are shown as mean + SD and results in (**e**) are shown as mean ± SD, *P* values were calculated by one-way ANOVA and Tukey's multiple comparisons test (\**P* < 0.05; \*\**P* < 0.01; \*\*\**P* < 0.001). Source data are provided as a Source Data file.

cells (Fig. 4d, e), whereas overexpression of *hda-1* or *dve-1* reduced the accumulation of polyQ aggregates in aged animals (Fig. 4f, g). Taken together, these results indicate that HDA-1 plays an important role in mediating the beneficial impact of UPR<sup>mt</sup> to promote lifespan extension in *C. elegans*. The data also suggest the therapeutic potential of activating HDAC signaling to manage certain age-related diseases, such as neurodegeneration.

**HDA-1 regulates mitochondrial stress response in mammals.** The beneficial effect of HDA-1 in modulating *C. elegans* mitochondrial homeostasis encouraged us to examine its physiological relevance in higher eukaryotes. HDAC1 forms homo- or heterodimers together with HDAC2 in various transcription regulator complexes such as NuRD and CoREST to repress or activate gene expression[39,40]. Interestingly, we found that the expression levels of HDAC1/2 strongly correlate with SATB2 (mammalian ortholog of DVE-1), the mitochondrial chaperones HSPA9 and HSPD1, the mitochondrial proteases LONP1 and

YME1L1, asparagine synthetase ASNS, and mitochondrial import inner membrane translocase TIMM17A in various human and rhesus monkey tissues (Fig. 5a; Supplementary Fig. 6). Furthermore, consistent with the interaction between HDA-1 and DVE-1 in *C. elegans*, HDAC1 also interacted with SATB2 in mammalian cells (Fig. 5b). Taken together, these results suggest that HDAC1/2 may play an evolutionarily conserved role in regulating mitochondrial homeostasis.

To validate the function of HDAC1/2 in mediating UPR<sup>mt</sup> during mitochondrial stress, we first treated HEK293T cells with sodium butyrate (NaBt), a chemical inhibitor of HDAC (Supplementary Fig. 7a). NaBt administration suppressed the induction of LONP1, HSPA9, YME1L1, and HSPD1 in cells treated with antimycin A, a well-known inhibitor of mitochondrial electron transport chain complex III (Fig. 6a; Supplementary Fig. 7b). However, it is possible that chemical inhibitors may indiscriminately target several HDACs, and the effect may not be specifically due to HDAC1/2 inhibition. To directly validate the role of HDAC1/2 in modulating mitochondrial

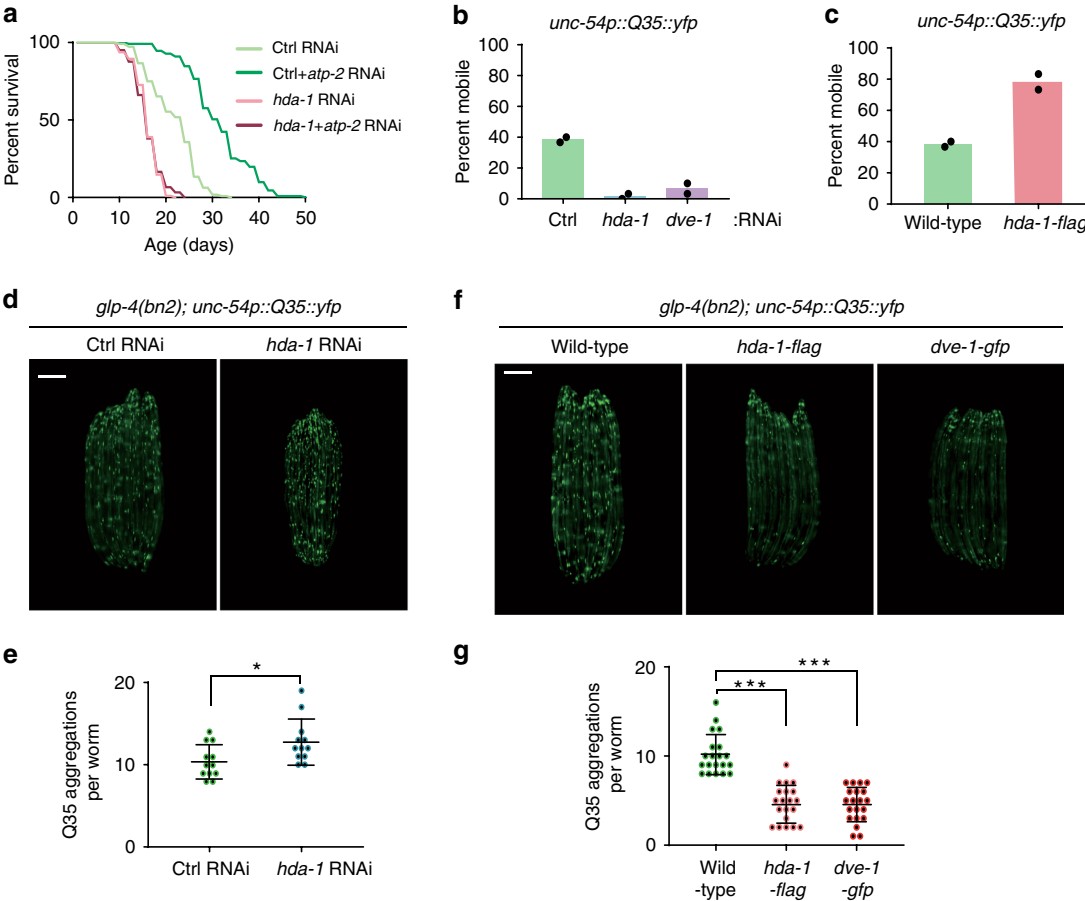

**Fig. 4 HDA-1 affects animal aging and age-dependent accumulation of protein aggregates. a** Lifespan analysis of worms raised on the indicated RNAis. $n = 130$ worms per condition. **b** Mobility analysis of *unc-54p::Q35::yfp* worms on control, *hda-1* or *dve-1* RNAi ($n = 2$ independent experiments, 30 worms per condition). **c** Mobility analysis of *unc-54p::Q35::yfp* and *unc-54p::Q35::yfp; hda-1p::hda-1::flag* worms on day 8 of adulthood ($n = 2$ independent experiments, 30 worms per condition). **d** Representative fluorescence images of *unc-54p::Q35::yfp* worms fed with control or *hda-1* RNAi. The images were taken on day 5 of adulthood. Scale bar, 200 μm. **e** Quantification of Q35 aggregates in (**d**). The numbers of Q35 aggregates in the body muscles (excluding the head and tail) of each worm were counted. $n = 12$ worms per condition. **f** Representative fluorescence images of *unc-54p::Q35::yfp*, *unc-54p::Q35::yfp; hda-1p::hda-1::flag* or *unc-54p::Q35::yfp; dve-1p::dve-1::gfp* worms on day 5 of adulthood. Scale bar, 200 μm. **g** Quantification of Q35 aggregates in (**f**). The numbers of Q35 aggregates in the body muscles (excluding the head and tail) of each worm were counted. $n = 20$ worms per sample. Results in (**e**, **g**) are shown as mean ± SD. In (**e**), the $P$ value was calculated by two-tailed Student's $t$ test (*$P < 0.05$). In (**g**), $P$ values were calculated by one-way ANOVA and Tukey's multiple comparisons test (***$P < 0.001$). Source data are provided as a Source Data file.

homeostasis and UPR^mt activation, we used siRNA to knock down HDAC1/2 or SATB2 in HeLa cells and stained mitochondria with MitoTracker. Compared to wild-type cells treated with antimycin A, disrupted mitochondrial morphology and higher mitochondrial fusion were observed in HDAC1/2- or SATB2-deficient cells challenged with antimycin A (Fig. 6b and Supplementary Fig. 7c). In addition, the induction of human UPR^mt genes such as HSPD1, LONP1, ASNS, and YME1L1 was significantly suppressed in HDAC1/2 or SATB2 knockdown cells challenged with mitochondrial insult (Fig. 6c and Supplementary Fig. 7d). Collectively, these results confirmed an evolutionarily conserved role of HDAC1/2 in mitochondrial surveillance and UPR^mt activation.

## Discussion
During mitochondrial stress, a mitochondrion-to-nucleus communication named UPR^mt is activated to initiate the transcriptional induction of genes involved in stress and immune responses, as well as those involved in metabolic reprogramming. Epigenetic regulation has emerged as another layer of regulation to relay mitochondrial stress signals to the expression of stress response genes[12–14,41]. Here we report that histone deacetylase HDA-1 coordinates with DVE-1 to activate UPR^mt in *C. elegans*. In addition, we provide evidence to show that its mammalian homologs HDAC1/2 play a conserved role to act in conjunction with SATB2 (mammalian ortholog of DVE-1) to mediate mitochondrial homeostasis.

Three-dimensional organization of chromatin architecture is important for regulating gene expression[42–44]. Interestingly, it has been well documented that SATB1 serves as a genome organizer to provide a landing platform for chromatin-remodeling enzymes such as HDAC[27,28,45]. Changes of chromatin organization such as the formation of chromatin loops will bring distant coregulated genes into close proximity and affect the accessibility of genomic loci. For instance, SATB1 has been reported to promote chromatin loop formation in T cells and activate cytokine gene expression[27]. Studies of HDAC1 have revealed that it can form chromatin-remodeling complexes such as NuRD and CoREST to regulate gene expression[39,40]. It will be important to examine the dynamic changes of chromatin architecture and characterize the function of HDA-1 and DVE-1 for rapid transcriptional induction in *C. elegans* upon mitochondrial perturbation. It will also be

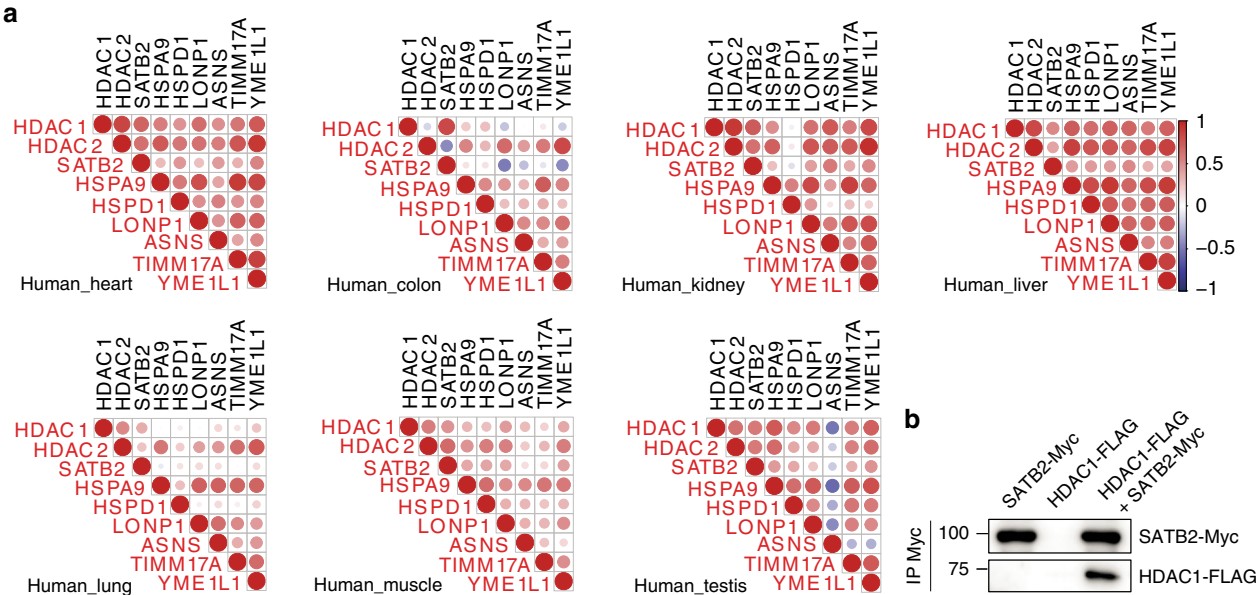

**Fig. 5 Transcript levels of human HDAC1/2 strongly correlate with the expression of UPR^mt genes. a** Pearson's correlation of HDAC1, HDAC2, SATB2 and UPR^mt mRNA levels in human heart, colon, kidney, liver, lung, muscle, and testis tissues. Red circles indicate positive correlation and blue circles indicate negative correlation. The size of the circle corresponds to the correlation coefficient. **b** Immunoprecipitation followed by immunoblotting shows that HDAC1 interacts with SATB2 in HEK293T cells. Source data are provided as a Source Data file.

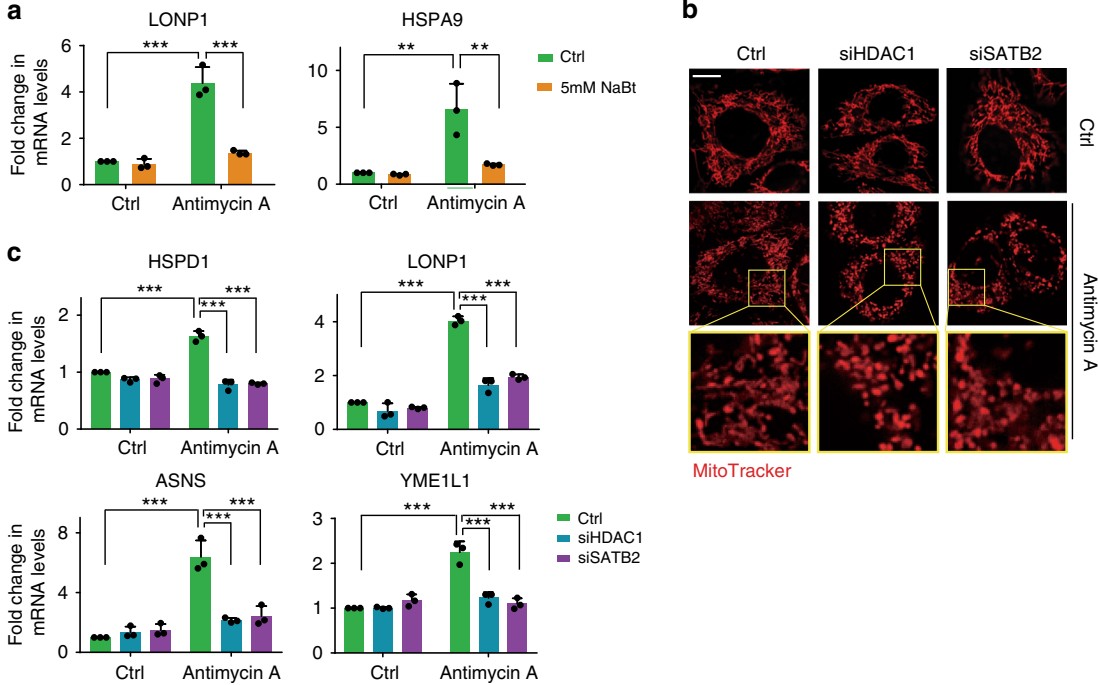

**Fig. 6 The functions of HDAC1/2 and SATB2 in UPR^mt activation are conserved in mammals. a** qRT-PCR measures mRNA levels of LONP1 and HSPA9 in HEK293T cells cultured under indicated conditions. **b** Representative fluorescence images of HeLa cells cultured under the indicated conditions. Scale bar, 10 μm. **c** qRT-PCR measures mRNA levels of UPR^mt genes in HEK293T cells cultured under the indicated conditions. (**a**, **c**) $n = 3$ independent experiments. Results in (**a**, **c**) are shown as mean + SD. $P$ values were calculated by two-way ANOVA and Tukey's multiple comparisons test (**$P < 0.01$; ***$P < 0.001$). Source data are provided as a Source Data file.

crucial to understand in the future how mitochondrial stress signals are relayed to HDA-1 activity.

Activation of UPR^mt not only upregulates the transcription of mitochondrion-specific chaperones and proteases to buffer the mitochondrial folding environment, but also initiates the transcriptional program to activate or suppress the expression of genes in different metabolic processes. Therefore, removal of the acetyl groups on histones through HDA-1 may indirectly adjust cellular metabolism to compensate for mitochondrial dysfunction by feeding additional acetyl-coA into metabolic reactions. Similarly, it has been shown that glucose or serum deprivation can reduce the levels of acetylated histones H3 and H4 in mammalian cells[46]. It will be of interest in the future to examine if global or specific acetylated histone marks are affected during mitochondrial perturbation.

Moderate mitochondrial stress can initiate a beneficial hormetic stress defense to promote innate immunity and lifespan extension[2,4,38,47,48], which suggests an intimate link between stress adaptation and the aging process[49]. Interestingly, we found that silencing of *hda-1* suppressed mitochondrial stress-induced immune response and longevity. More importantly, over-expression of *hda-1* delayed age-related accumulation of protein aggregates and polyglutamine toxicity in *C. elegans*. In addition, systematic correlation analysis in rhesus monkey and human strongly points to a conserved function of HDAC1/2 in regulating mitochondrial hormesis in primates. Therefore, future genetic or pharmacological treatments to target this pathway may provide therapeutic potential for the treatment of age-related diseases.

## Methods

**Strains and culture**. SJ4100(zcIs13[hsp-6::gfp]), SJ4058(zcIs9[hsp-60p::gfp]), CL2070 (dvIs70[hsp-16.2::gfp]), SJ4005(zcIs4[hsp-4p::gfp]), CB5535(hda-1 (e1795)), SJ4197(zcIs39[dve-1p::dve-1::gfp]), AU133(agIs17[irg-1p::gfp]), AM140 (rmIs132 [unc-54p::Q35::yfp]), SS104(glp-4(bn2)), and N2 wild-type C. elegans were obtained from the Caenorhabditis Genetics Center.

The following strains were generated in our lab: YSL1(liuls1[hda-1p::hda-1::flag; odr-1p::dsRed]), YSL2(liuls2[hda-1p::hda-1::gfp; odr-1p::dsRed]), YSL3(liuls1[hda-1p::hda-1::flag; odr-1p::dsRed]; zcIs39[dve-1p::dve-1::gfp]), YSL4(zcIs39[dve-1p:: dve-1::gfp]; liuEx1[hda-1p::hda-1::mCherry; odr-1p::dsRed]), YSL5(liuls1[hda-1p:: hda-1::flag; odr-1p::dsRed]; zcIs13[hsp-6p::gfp]), YSL6(glp-4(bn2); liuls2[hda-1p:: hda-1::gfp; odr-1p::dsRed]), YSL7(glp-4(bn2); zcIs39[dve-1p::dve-1::gfp]), YSL8 (glp-4(bn2); rmIs132[unc-54p::Q35::yfp]), YSL9(glp-4(bn2); liuls1[hda-1p::hda-1:: flag; odr-1p::dsRed]; rmIs132 [unc-54p::Q35::yfp]), YSL10(glp-4(bn2); zcIs39[dve-1p::dve-1::gfp]; rmIs132 [unc-54p::Q35::yfp]), and YSL11(glp-4(bn2); liuEx2[gly-19p::hmgs-1::flag; mec-7p::rfp]).

HEK293T cells and HeLa cells were obtained from ATCC. Cells were cultured in DMEM medium supplemented with 10% (v/v) fetal bovine serum at 37 °C. SATB2 plasmid was constructed by PCR amplification of cDNA from total HEK293T RNA and ligated into the pCDNA3.3 vector. HDAC1 plasmid was a gift from Prof. Jiemin Wong. Plasmids were transfected into cells by Lipofectamine 2000 Reagent.

Primer sequences for plasmid construction are provided in Supplementary Table 1.

**RNA interference**. Most RNAi clones were obtained from the Ahringer library, whereas *hda-3* and *hda-5* RNAi clones were generated by PCR amplification of worm cDNA and ligated into L4440 vector. The clones were then transformed into HT115 competent cells. RNAi clones were grown in LB at 37 °C overnight. 20X concentrated bacterial culture solution was seeded on to worm plates with 1.2 mg/ml IPTG. Dried plates were kept at room temperature overnight to allow IPTG induction of double-stranded RNA (dsRNA) expression. Synchronized L1 worms were raised on RNAi plates at 20 °C. For double RNAi experiments, mixed bacterial culture solutions were seeded onto RNAi plates, or the second RNAi bacterial culture solution was seeded 30 h after worms were cultured on the first RNAi clones.

For siRNA knockdown in mammalian cells, 50 pmol of siRNA was transfected into a 6-well plate by Lipofectamine RNAi MAX Reagent.

**Induction of UPR^mt^**. For induction of UPR^mt^ by RNAi, synchronized L1 worms raised on *atp-2* or *cco-1* RNAi were imaged when they reached adulthood. For double RNAi experiments, 20X concentrated *atp-2* or *cco-1* RNAi bacterial culture (pre-induced by 0.2 mg/ml IPTG) was seeded 30 h after worms were cultured on the first RNAi. After another 30 h, worms were imaged.

For induction of UPR^mt^ by antimycin A in cells, HEK293T cells with 80% confluency were treated with 20 μg/ml antimycin A (Sigma #A8674) for 21 h.

**Induction of UPR^ER^ and HSR**. For induction of UPR^ER^ by tunicamycin, synchronized L1 worms were raised on 6 cm RNAi plates for 43 h. 300 μl M9 buffer containing 18 μg tunicamycin was then spread across the entire surface of the plate. Induction of UPR^ER^ was examined after 12 h.

To induce the heat shock response (HSR), synchronized L1 worms were raised on 6 cm RNAi plates for 34 h, cultured at 37 °C for 1 h and then transferred to 20 °C for another 21 h before examining the induction of HSR.

**Induction of innate immune response**. For induction of the innate immune response by pathogens, synchronized L1 worms were raised for 24 h on worm plates before exposure to *P. aeruginosa* or *Rhodococcus*. Worms were imaged on day 2 of adulthood.

**Sodium butyrate treatment**. Cells were treated with 5 mM sodium butyrate (Sigma #303410) for 24 h. For sodium butyrate and Antimycin A double treatment, Antimycin A was added into the cell culture medium 3 h after sodium butyrate treatment.

**Microscopy**. Worms were picked into 100 mM NaN₃ droplets on 2% agarose pads and imaged by a Zeiss Imager M2 microscope. Nuclear localization of worms was imaged by a Zeiss LSM880 microscope. Mitochondrial morphology of HeLa cells was imaged by a PerkinElmer Operetta CLS™. Comparable images were captured with the same exposure time and magnification. GFP fluorescence was quantified by ImageJ. For imaging mitochondria, HeLa cells were treated with 50 μg/ml antimycin A (Sigma #A8674) for 4 h and then stained with MitoTracker (Invitrogen #M7512).

**RNA isolation and real-time PCR**. Synchronized late L4 worms or HEK293T cells were collected and resuspended with trizol reagent (Cwbiol #cw0580A). Worm samples were frozen and homogenized three times in liquid nitrogen. Total RNA was isolated by chloroform extraction, precipitated with isopropanol, then washed with 75% (v/v) ethanol. cDNA was synthesized with reverse transcription kits (Transgen #AT311). Quantitative PCR was carried out using SYBR Green PCR Master Mix (Bio-Rad #1725121). For quantification, *C. elegans* transcripts were normalized to *rpl-32*, and transcripts from HEK293T cells were normalized to ACTB. Primer sequences for quantitative RT-PCR are provided in Supplementary Table 2.

**Immunoblotting**. Worms or cells were resuspended with SDS loading buffer (100 mM Tris-HCl pH 6.8, 4% SDS, 20% glycerol, 10% β-mercaptoethanol, 0.004% bromophenol blue) and boiled at 95 °C for 10 min. Samples were separated by SDS-PAGE and transferred onto a PVDF membrane (Bio-Rad). After blocking with 5% milk-TBST, the membrane was probed with the designated primary antibodies and secondary antibodies. Primary antibodies used in this study: anti-GFP (Sungene #KM8009, 1:1000 dilution), anti-tubulin (Abcam #ab6161, 1:1000 dilution), anti-Myc (CST #2276, 1:1000 dilution), anti-FLAG (Sigma #F7425, 1:2000 dilution) and anti-Acetylated-Lysine (CST #9441, 1:1000 dilution). Membranes were developed with the enhanced chemiluminescence method (Thermo) and visualized using the Tanon 5200 chemical luminescence imaging system.

**Immunoprecipitation**. For worm samples, 30,000 young adult worms with the indicated genotype were washed off by M9 buffer and resuspended in 3 ml lysis buffer (50 mM tris-HCl pH 8.0, 137 mM NaCl, 1% Triton X-100, 1 mM EDTA, 10% glycerol, proteinase inhibitor). Samples were homogenized with a glass homogenizer and sonicated. For HEK293T cells, cells in one 10 cm dish with 90% confluence were washed with 1X PBS buffer, then resuspended in 1 ml lysis buffer and sat on ice for 30 min. The worm or cell lysate was centrifuged at 20,000 × *g* for 15 min. The supernatant was then transferred into a new tube and rotated at 4 °C overnight in the presence of the designated antibody anti-GFP antibody (Abcam #ab290, 1 μl per sample), anti-FLAG magnetic beads (Sigma #M8823, 40 μl per sample), anti-Myc magnetic beads (Bimake #B26302, 20 μl per sample). For anti-GFP immunoprecipitation, protein G beads (Invitrogen #10004D, 40 μl per sample) were subsequently added to each sample and rotated at 4 °C for additional 2 h. After binding, the beads were washed three times with lysis buffer and boiled in 50 μl 2X SDS Laemmli buffer (4% SDS, 20% glycerol, 10% 2-mercaptoethanol, 0.02% bromophenol blue, 0.125 M Tri-HCL, pH 6.8) at 95 °C for 10 min.

**Lifespan measurement**. Synchronized L1 worms were raised on fresh plates seeded with the indicated bacteria. After reaching adulthood, worms were transferred to fresh plates every 2 days and monitored for survival. A worm that did not respond to three gentle touches on the head and displayed no pharyngeal pumping was considered dead. Those that died due to internal hatching, ruptured vulvae, or crawling off the agar were removed.

**P. aeruginosa slow-killing assay**. *P. aeruginosa* was cultured overnight in LB containing 50 μg/ml kanamycin at 37 °C. To prepare *P. aeruginosa* plates, 10 μl *P. aeruginosa* was seeded onto 3.5 cm slow-killing agar plates[50] and spread slightly to a small circle using a sterile L spreader. Plates were air-dried, incubated at 37 °C for 24 h and then incubated at room temperature for another 24 h. Synchronized L1 *glp-4(bn2)* worms were first raised on RNAi plates at 20 °C for 49 h, then 50 worms were randomly picked onto *P. aeruginosa* plates and cultured at 25 °C. To score survival rate of N2 wild-type worms, 5-fluoro-2′-deoxyuridine (FUdR) (100 μg/ml) was added in the *P. aeruginosa* plates. Living worms were counted every 24 h. Those crawling off the agar were removed.

**Rhodococcus survival assay**. *Rhodococcus* was cultured overnight in LB at 25 °C. A 30X concentrated solution of bacteria was seeded on to 6 cm slow-killing agar plates. Synchronized L1 *glp-4(bn2)* worms were first raised on RNAi plates at 20 °C for 49 h, then about 50 worms were randomly picked onto dried *Rhodococcus* plates and cultured at 25 °C. Living worms were counted every 24 h. Those crawling off the agar were removed.

**P. aeruginosa intestinal accumulation assay**. The *Pseudomonas aeruginosa* strain PA01 with *pMF230* (Addgene_62546), which expresses GFP, was provided by Dr. Huanqin Dai. To prepare PA01(GFP) plates, overnight cultures of PA01(GFP) were spread across the entire surface of the slow-killing plates. Plates were air-dried, incubated at 37 °C for 24 h and then incubated at room temperature for 24 h. Synchronized worms fed with RNAi at 20 °C for 53 h were washed twice in M9 buffer and transferred onto PA01(GFP) plates. Worms were imaged after another 45 h.

**Quantification of PA01(GFP) colony-forming units (CFU)**. Worms were exposed to PA01(GFP) in the same conditions as in the *P. aeruginosa* intestinal accumulation assay. After a defined period, worms were washed three times in M9 buffer and rotated for 1 h in M9 buffer containing kanamycin (1 mg/ml). After three additional washes with M9 buffer, 30 worms were picked into 100 μl M9 buffer in a 1.5 ml tube and grinded with a motorized pestle. Lysates were serially diluted, and 5 μl lysate solution was plated on LB plates containing carbenicillin (50 μg/ml). After overnight incubation at 37 °C, colonies with GFP fluorescence were counted.

**Mobility assay**. On day 8 of adulthood, worms with Q35 polyglutamine expression were touched two times on the head and tail. Animals which could move and change their physical position were counted. Worms were randomly selected and tested under each condition.

**ChIP and ChIP-seq data analysis**. For each ChIP sample, 240,000 worms with HDA-1-GFP or DVE-1-GFP overexpression were harvested. After cross-linking with 2% formaldehyde for 40 min and washing 3 times by cold PBS, worm pellets were resuspended in FA buffer (50 mM HEPES/KOH pH 7.5, 1 mM EDTA pH 8.0, 1% Triton-X-100, 0.1% sodium deoxycholate) with 150 mM NaCl and proteinase inhibitor, then homogenized, and sonicated. Next, the samples were centrifuged at $20,000 \times g$ for 20 min at 4 °C. After dilution with FA buffer (containing 150 mM NaCl), 1% (V/V) supernatant was kept as the Input, and the rest was incubated with GFP-Trap agarose (ChromoTek #gta-20, 50 μl per sample) at 4 °C for 20 h. Then the GFP-Trap agarose was washed twice with 150 mM NaCl FA buffer, once with 1 M NaCl FA buffer, twice with 500 mM NaCl FA buffer, once with LiCl buffer (10 mM Tris-HCl pH 8.0, 1 mM EDTA pH 8.0, 1% NP40, 1% sodium deoxycholate, 250 mM LiCl) and three times with TE buffer. The bound fraction was eluted from the agarose by TE buffer (containing 1% SDS, 250 mM NaCl) and the elute was decross-linked together with the Input overnight at 65 °C. The samples were then digested by protease K for 2 h at 55 °C[51]. DNA was extracted with a ChIP DNA Clean & Concentrator kit (Zymo Research #D5201) and DNA libraries were prepared using an NEB kit (NEB # E7370). For ChIP-seq data analysis, adaptor sequences were filtered out of reads using an in-house script, then Trim Galore was used to do the quality control with the following parameters: -q 20 --fastqc --illumina --stringency 6 -e 0 --length 50 --trim-n --paired. Processed reads were mapped to the *C. elegans* genome (ce11) using BWA (version 0.7.13-r1126)[52] with default parameters. Only uniquely mapping reads that were properly paired with mapping quality ≥20 and mismatches <8 were retained. PCR duplicates were removed by Picard (version 2.17.6) (http://broadinstitute.github.io/picard). MACS2 (version 2.1.1)[53] was used to obtain peak regions with the following parameters: -g ce -f BAMPE --broad --SPMR. ChIP-seq signals were calculated using deepTools (version 2.4.2)[54] by normalizing the read coverage to 10 million reads then subtracting input coverage from the ChIP samples to get the final ChIP-seq signals. Overlaps between peaks were determined by the BEDTools (version 2.27.1)[55] intersect command. Motif analysis was performed by HOMER (version 4.10-0)[56] with the following parameters: ce11 -gc -size given. Bwtool[57] was used for peak signal comparison between groups.

**mRNA-seq sample preparation and data analysis**. For worm samples, about 1000 synchronized L1 worms were fed with control, *hda-1* RNAi or *dve-1* RNAi for 24 h, followed by *atp-2* RNAi for 48 h. Worms were washed and resuspended with trizol reagent for total RNA isolation.

Reads were aligned to the *C. elegans* genome (ce11) using TopHat2 (version 2.1.1)[58] with the following parameters: --read-mismatches 6 --read-edit-dist 6 --min-anchor 8 --segment-length 26. Uniquely mapping reads were used for further analysis. Gene expression levels and differentially expressed genes were generated by Cufflinks (version 2.2.1)[59]. GO term enrichment analysis was performed by DAVID[60,61] using GOTERM_BP_2 category.

**Expression correlation analyses**. Correlation analysis of the expression levels of HDAC1, HDAC2, SATB2 and UPR^mt genes in different human tissues was performed using data from GTEx (GTEx v7 All Tissues RNA-Seq, https://www.gtexportal.org/home/datasets). Correlation analysis of the expression levels of HDAC1, HDAC2, SATB2, and UPR^mt genes in different rhesus monkey tissues was performed using data from RhesusBase[62] (http://rhesusbase.cbi.pku.edu.cn/download/download.jsp). In each tissue, the Pearson's correlation for each pair of genes was calculated using the gene expression levels among all samples in human or rhesus macaque stated above.

**Statistics and reproducibility**. Statistical analyses were performed with GraphPad Prism 5.0. Results were expressed as mean ± SD or mean + SD. ANOVA and Tukey's multiple comparisons test or Student's *t* test was used to calculate the *P* values. For details of the particular statistical analyses employed, precise *P* values and statistical significance for all graphs, see figure legends. Experiments yielding quantitative data for statistical analysis were performed independently at least twice, all with similar results. Micrographs and immunoblotting images shown in the figures are representative of three independent experiments, all with similar results.

**Reporting summary**. Further information on research design is available in the Nature Research Reporting Summary linked to this article.

## Data availability

Sequencing data reported in this paper have been deposited in the NCBI Gene Expression Omnibus (GEO) database under accession codes GSE141041 (RNA-seq of worms), GSE141042 (ChIP-seq of worms). Gene expression levels in different human tissues were obtained from https://www.gtexportal.org/home/datasets. Gene expression levels in different rhesus monkey tissues were obtained from http://rhesusbase.cbi.pku.edu.cn/download/download.jsp. The source data underlying Figs. 1b, c, 2a, 3b, c, e, f, h, 4a–c, e, g, 5a, 6a, c and Supplementary Figs. 1a, b, d, f, h, 2a, c, d, 3a, b, 4a–c, e–g, j, 5a–c, 6, 7a, b, d are provided as a Source data file. Other data supporting the findings of this study are available within the paper and the Supplementary Information files, or available from the authors upon reasonable request. Source data are provided with this paper.

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

## Acknowledgements

We thank the *Caenorhabditis* Genetics Center for providing strains. We are grateful to Dr. Isabel Hanson for editing the manuscript. Y.L. was supported by grants from the National Natural Science Foundation of China (grants no.91854205 and 31925012), the Ministry of Science and Technology of China (National Key Research and Development Program of China grant no.2017YFA0504000973), and an HHMI International Research Scholar Program (grant no. 55008739). This work was also supported by Beijing Advanced Innovation Center for Genomics and Peking-Tsinghua Center for Life Sciences. L.-W. Shao was supported by National Natural Science Foundation of China (grants no. 31900544) and the China Postdoctoral Science Foundation (grant no. 2018M640021). C.-Y.L. was supported by grants from the Ministry of Science and Technology of China (National Key Research and Development Program of China grant no.2019YFA0801801 and no.2018YFA0801405) and the National Natural Science Foundation of China (grant no.31871272). Y.L. acknowledges the support from the Tencent Foundation through the XPLORER PRIZE.

## Author contributions

L.-W.S. and Y.L. conceived the study. M.D. carried out RNAi and pharmacological inhibition experiments in mammalian cells. K.G. performed immunoprecipitation in mammalian cells and carried out some lifespan analysis and the *P. aeruginosa* slow-killing assay in worms. Y.L. carried out some micro-injection experiments to generate transgenic worm strains. L.-W. Shao performed the rest of the experiments. Q.P. and Y.L. performed bioinformatics analyses under the supervision of C.-Y.L. L.-W.S., Q.P., C.-Y.L., and Y.L. wrote the manuscript.

## Competing interests

The authors declare no competing interests.
