## [Peer Review File · Nature Communications]

Reviewers' comments, first round:

Reviewer #1 (Remarks to the Author):

The manuscript by Shao et al entitled “Histone deacetylase HDA-1 modulates mitochondrial stress response and longevity” evaluated the mechanisms by which HAD-1 regulates mitochondrial stress. They identified *hda-1* from a genome-wide RNAi screen for genes that modulate *atp-2* RNAi-induced upregulation of *hsp6::gfp*, which is a UPRmt reporter. They then identified *atp-2* induced genes that are dependent on *hda-1* through RNA-seq. These *hda-1*-dependent genes are enriched for genes involved in immune response and stress response. They further uncovered DVE-1 as a cofactor from proteomic analysis and showed that HDA-1 and DVE-1 co-regulated a set of immune response genes through CHIP-seq. They then showed that HDA-1 was required for UPRmt-mediated innate immunity and lifespan extension, and played a role in longevity and age-dependent protein aggregation. Lastly, they expanded their findings to mammalian cells and suggested that the role of HDA-1 in mitochondrial stress response is evolutionarily conserved. Overall, I found the genetics experiments to be quite compelling and comprehensive, but the part of mechanistic dissection can be improved.

1. Are there specific HDA-1 target genes that mediate HDA-1 function? They identified 283 HDA-1-dependent genes and 139 genes co-regulated by HDA-1 and DVE-1. They could validate the expression regulation with qRT-PCR for some key genes and manipulate these genes to test whether these target genes mediate, at least partially, the role of HDA-1.

2. They found that HDA-1 and DVE-1 co-bound to a set of genes. What are the binding motifs for the two proteins? Does the binding depend on each other?

Minor concerns:

1. They showed that *hda-1* RNAi shortened lifespan (Fig. 4a). However, it has been previously shown that *hda-1* mutant does not affect longevity [Aging (Albany NY). 2014 Aug; 6(8): 621–644]. They should at least discuss about it.

2. Lifespan curves in Fig. 4a are in colors that are really close to each other, making it very hard to read the graph.

3. RNAi is a powerful approach that is extensively used in the manuscript. But it will be valuable to confirm some of the key data with genetic mutations.

Reviewer #2 (Remarks to the Author):

The manuscript by Shao, Peng et al. identify the *C. elegans* histone deacetylase HDA-1 as a regulator of the mitochondrial unfolded protein response (UPRmt). HDA-1 was among the only histone deacetylases tested that was required for UPRmt activation and not for other cellular stress responses. HDA-1 and the SATB2 homolog DVE-1 were found to interact with each other during the activation of the UPRmt. Both HDA-1 and DVE-1 function were required for protection against

infection and the increased longevity observed with UPRmt activation. Finally, HDA-1 appears to have a conserved role in the activation of the UPRmt since transcript levels of its human homolog HDAC1/2 were correlated with known UPRmt genes. And, along with the DVE-1 human homolog SATB2, HDAC1 was required for their induction with mitochondrial stress.

Overall, the manuscript submitted by Shao, Peng et al. is a well-written and thoroughly performed study which identifies a new mode of regulation for the UPRmt. However, the idea that chromatin remodelers are involved in the regulation of the UPRmt and longevity has been observed before (see PMID 27133166 and 27133168). Nonetheless, the following issues should be addressed.

Major revisions

Figure 2b: HDA-1 and DVE-1 expression appears to rely on each other. The authors suggest that this is due to a ubiquitin-mediated degradation mechanism since loss of the ubiquitin gene *ubq-1* and *ubq-2* restore expression in the absence of each partner. However this may still represent an indirect effect. Alternatively, is the reduction in GFP expression due to reduced transcription of each gene? Employing transcriptional GFP reporters of *hda-1/dve-1* may help resolve this, or simply performing qPCR of *hda-1* or *dve-1* in the presence or absence of the respective partner.

Also regarding Figure 2b: the difference in HDA-1::GFP expression between control and *dve-1* RNAi is not very impressive. Some form of quantification is required.

Figure 2f and 2g: HDA-1 and DVE-1 were found to bind a considerable number of genes in the absence of stress by ChIP. What is the overlap between HDA-1 and DVE-1 regulated genes in the absence of stress by RNAseq?

Also, what number of stress-activated HDA-1/DVE-1 genes are in common between the ChIPseq and RNAseq in Figure 2f and g?

Figure 3a: The authors use the gene *irg-1* as a readout for immune response induction. However, *irg-1* was shown to be responsive to translation inhibition and itself is not needed for protection during infection. However, the authors have uncovered a number of immune response genes whose expressions are dependent on both HDA-1 and DVE-1 (Figure 2h). Are these immune response genes induced during infection in a HDA-1/DVE-1-dependent manner (a subset of these genes would suffice)?

Figure 3c: HDA-1 overexpression was found to extend host survival during infection with *P. aeruginosa* infection. Does HDA-1 overexpression also reduce pathogen colonization?

How many independent transgenic lines were tested?

Does DVE-1 overexpression also provide protection during infection?

Figure 3d: *Pseudomonas aeruginosa*-GFP was used as a readout for colonization levels following *hda-1/dve-1* RNAi. This assay requires some form of quantification since *P. aeruginosa* colonization can be quite variable. Ideally, one would perform colony forming unit quantifications of worm lysates to accurately quantify pathogen accumulation.

Figure 3a and 3e: Can the authors please comment on why *irg-1::GFP* expression is fully dependent on DVE-1 during *P. aeruginosa* infection but is only partially dependent during *Rhodococcus* infection whereas HDA-1 is fully required for both?

Also, is HDA-1 and DVE-1 required for survival during *Rhodococcus* infection?

Figure 4a: The authors show that *hda-1* RNAi completely suppresses the increase in lifespan of *atp-2* RNAi animals. However, *hda-1* RNAi also accelerates worm mortality by itself. Is the effect of *hda-1* RNAi on *atp-2* RNAi animal lifespan therefore specific in terms of mitochondrial stress induced longevity? In this regard, does *hda-1* RNAi suppress other long-lived animals associated with other longevity pathways (e.g. caloric restriction/*eat-2* etc.)?

Also, HDA-1 overexpression was able to increase the survival of *C. elegans* during infection. Is HDA-1 overexpression sufficient to increase longevity as well?

Figure 4d: The authors claim that HDA-1 overexpression reduces polyQ aggregation, however the differences are not overtly obvious based on the images presented. Again, some quantification of the images is required. Ideally, one could compliment their visualization of Q35::YFP aggregation by testing biochemically for changes in poly Q solubility by SDS-PAGE and Western Blotting using worm lysates. And, does DVE-1 overexpression also reduce Q35 aggregation similar to HDA-1?

Also, the authors only test for the effect of HDA-1 overexpression on protein aggregation (i.e. *hda-1* RNAi is only perform for the mobility assay). Is the converse true for *hda-1* RNAi?

For Figure 6b: the differences in mitochondrial morphology are not obvious in the presence or absence of HDAC1 or SATB2 during stress. Also, the authors perform all qPCRs using HEK293 cells but the morphology was performed using Hela cells. Is there a reason for the discrepancy?

Minor revisions

- Figure 2b: red arrows are not referenced in figure legend
- Figure 6b: The authors state in the Results section that HDAC2 loss had similar effects on mitochondrial morphology but HDAC2 is not represented in the actual figure.
- Supplementary Figure 2b: the differences in DVE-1::GFP expression is not obvious. Quantifications are required.
- The authors reference to Supplementary Figure 6 in the results section which does not exist.

Reviewers' comments:

Reviewer #1 (Remarks to the Author):

The manuscript by Shao et al entitled “Histone deacetylase HDA-1 modulates mitochondrial stress response and longevity” evaluated the mechanisms by which HAD-1 regulates mitochondrial stress. They identified *hda-1* from a genome-wide RNAi screen for genes that modulate *atp-2* RNAi-induced upregulation of *hsp-6::gfp*, which is a UPRmt reporter. They then identified *atp-2* induced genes that are dependent on *hda-1* through RNA-seq. These *hda-1*-dependent genes are enriched for genes involved in immune response and stress response. They further uncovered DVE-1 as a cofactor from proteomic analysis and showed that HDA-1 and DVE-1 co-regulated a set of immune response genes through ChIP-seq. They then showed that HDA-1 was required for UPRmt-mediated innate immunity and lifespan extension, and played a role in longevity and age-dependent protein aggregation. Lastly, they expanded their findings to mammalian cells and suggested that the role of HDA-1 in mitochondrial stress response is evolutionarily conserved. Overall, I found the genetics experiments to be quite compelling and comprehensive, but the part of mechanistic dissection can be improved.

We appreciate the reviewer's interest in our work and his/her thoughtful consideration of our manuscript.

1. Are there specific HDA-1 target genes that mediate HDA-1 function? They identified 283 HDA-1-dependent genes and 139 genes co-regulated by HDA-1 and DVE-1. They could validate the expression regulation with qRT-PCR for some key genes and manipulate these genes to test whether these target genes mediate, at least partially, the role of HDA-1.

Following the reviewer's suggestion, we carried out qRT-PCR to validate HDA-1- and DVE-1-regulated expression of key genes associated with several of the most strongly enriched GO terms. Knockdown of *hda-1* or *dve-1* significantly suppressed the induction of *cdr-4*, *pgp-1*, *nhr-115*, *M04C3.2* (GO term “immune response”), *dnj-10*, *djr-1.2* (GO term “stress response”), *cyp-33C8*, *ipla-3*, *hmgs-1*, *tars-1* (GO term “single-organism metabolic process”) and *cua-1* (GO term “single-organism localization”). We have provided these results in Supplementary Fig. 3a of the revised manuscript.

We also followed the reviewer's suggestion to manipulate some of the key genes to test if the target gene partially mediates the function of HDA-1. From the 139 genes co-regulated by HDA-1 and DVE-1, we tested 23 genes with functional annotation, including the genes associated with the most strongly enriched GO terms that are mentioned above. We found

that knockdown of *hmgs-1* suppressed the immune response, resulting in the reduced survival rate of worms exposed to *P. aeruginosa* and the accumulation of *P. aeruginosa* (GFP) (Supplementary Fig. 4g, h). *hmgs-1*, one of the 139 genes co-regulated by HDA-1 and DVE-1 (Supplementary Fig. 3 and Supplementary Table 1), encodes an HMG-CoA synthase. We then tried to overexpress *hmgs-1* in *hda-1* RNAi worms. However, overexpression of *hmgs-1* was not able to suppress the accumulation of *P. aeruginosa* (GFP) or rescue the survival rate of *hda-1* RNAi worms exposed to *P. aeruginosa* (Supplementary Fig. 4i, j). These results suggest that expression of this target gene alone is not sufficient to rescue *hda-1*-deficient phenotypes.

2. They found that HDA-1 and DVE-1 co-bound to a set of genes. What are the binding motifs for the two proteins?

We analyzed the HDA-1 and DVE-1 ChIP-seq results, and revealed the binding motifs of these two proteins in the presence or absence of mitochondrial stress (Figure 2g).

Does the binding depend on each other?

We thank the reviewer for raising this question. Our ChIP-seq analysis indicated that under mitochondrial stress, the upregulated signals of HDA-1 ChIP-seq peak were suppressed by knockdown of *dve-1* (Fig. 2h). Conversely, knockdown of *hda-1* did not affect *dve-1* ChIP signals (Fig. 2i). These results suggest that the binding of HDA-1 depends on DVE-1. It is also in agreement with a study in mammals, which reported that SATB1, the DVE-1 homolog in higher eukaryotes, can provide a docking site to recruit histone deacetylase HDAC1 onto SATB1 target sequences (PMID: 12374985). We included the new analyses and discussions in the revised manuscript (last paragraph, Page 7).

Minor concerns:

1. They showed that *hda-1* RNAi shortened lifespan (Fig. 4a). However, it has been

previously shown that *hda-1* mutant does not affect longevity [Aging (Albany NY). 2014 Aug; 6(8): 621–644]. They should at least discuss about it.

We thank the reviewer for the insightful suggestion. The referenced study (PMID: 25127866) by Edwards *et al.* also used *hda-1* RNAi, instead of an *hda-1* mutant, to perform lifespan analysis. We think the difference may due to two reasons:

- 1) After addition of IPTG to induce dsRNA expression, Edwards *et al.* only allowed 4 hours of induction, where we followed a standard protocol to allow for an overnight induction. We speculate that the *hda-1* knockdown efficiency is higher in our study. We have achieved ~80% knockdown (Supplementary Fig. 5a).
- 2) Edwards *et al.* employed a liquid culture system to study lifespan, whereas we cultured worms and performed lifespan analysis using agar plates.

Intrigued by the reviewer's question, we also tested the lifespan of worms carrying a *hda-1(e1795)* hypomorphic allele and found that the *hda-1* mutant also had a shortened lifespan (Supplementary Fig. 5b). Followed the reviewer's suggestions, we included more discussions in the revised manuscript to address this issue (Page 11).

2. Lifespan curves in Fig. 4a are in colors that are really close to each other, making it very hard to read the graph.

We have followed the reviewer's suggestion to change the colors in Fig. 4a.

3. RNAi is a powerful approach that is extensively used in the manuscript. But it will be valuable to confirm some of the key data with genetic mutations.

We appreciate this point from the reviewer. In the revised manuscript, we added substantial new data to confirm some of our key findings with an *hda-1(e1795)* hypomorphic allele. Worms carrying the *e1795* allele, containing an amino acid substitution, are sterile and die prematurely. We confirmed that the *hda-1(e1795)* mutation also suppressed the induction of mitochondrial stress response genes in worms treated with *atp-2* RNAi (Supplementary Fig. 3b). In addition, the *hda-1(e1795)* mutation resulted in the accumulation of *P. aeruginosa* and reduced the survival rate of worms when they were exposed to *Pseudomonas* (Supplementary Fig. 4c, d).

Reviewer #2 (Remarks to the Author):

The manuscript by Shao, Peng et al. identify the *C. elegans* histone deacetylase HDA-1 as a regulator of the mitochondrial unfolded protein response (UPR_{mt}). HDA-1 was among the

only histone deacetylases tested that was required for UPR^{mt} activation and not for other cellular stress responses. HDA-1 and the SATB2 homolog DVE-1 were found to interact with each other during the activation of the UPR^{mt}. Both HDA-1 and DVE-1 function were required for protection against infection and the increased longevity observed with UPR^{mt} activation. Finally, HDA-1 appears to have a conserved role in the activation of the UPR^{mt} since transcript levels of its human homolog HDAC1/2 were correlated with known UPR^{mt} genes. And, along with the DVE-1 human homolog SATB2, HDAC1 was required for their induction with mitochondrial stress.

Overall, the manuscript submitted by Shao, Peng et al. is a well-written and thoroughly performed study which identifies a new mode of regulation for the UPR^{mt}. However, the idea that chromatin remodelers are involved in the regulation of the UPR^{mt} and longevity has been observed before (see PMID 27133166 and 27133168). Nonetheless, the following issues should be addressed.

We thank the reviewer for the encouraging remarks on our work. We would like to thank the reviewer for the careful review of our manuscript. We also agree with the reviewer that the two reference studies published in *Cell* (PMID 27133166 and 27133168) indicate that histone methylation is involved in the regulation of the UPR^{mt}. However, the identification of HDA-1 in our study suggests for the first time that histone deacetylation also occurs during UPR^{mt} activation. In addition, based on the analyses in cells and tissues of both rhesus monkey and human, we further revealed that HDAC/HDA-1 and SATB1/DVE-1 play an evolutionarily conserved role to protect mitochondrial function. Therefore, our study adds further mechanistic and evolutionary insights into the chromatin remodeling events that occur during the global induction of stress response and immune response genes.

Major revisions

1. Figure 2b: HDA-1 and DVE-1 expression appears to rely on each other. The authors suggest that this is due to a ubiquitin-mediated degradation mechanism since loss of the ubiquitin gene *ubq-1* and *ubq-2* restore expression in the absence of each partner. However, this may still represent an indirect effect. Alternatively, is the reduction in GFP expression due to reduced transcription of each gene? Employing transcriptional GFP reporters of *hda-1/dve-1* may help resolve this, or simply performing qPCR of *hda-1* or *dve-1* in the presence or absence of the respective partner.

We followed the reviewer's suggestion to carry out qPCR analysis of *hda-1* or *dve-1* transcription levels in the presence or absence of the respective partner. We showed that the transcription of *hda-1* or *dve-1* is not affected by the level of its partner (Supplementary Fig. 2d).

2. Also regarding Figure 2b: the difference in HDA-1::GFP expression between control and *dve-1* RNAi is not very impressive. Some form of quantification is required.

We have followed the reviewer's suggestion and quantified the HDA-1::GFP expression level by immunoblotting (Supplementary Fig. 2a).

3. Figure 2f and 2g: HDA-1 and DVE-1 were found to bind a considerable number of genes in the absence of stress by ChIP. What is the overlap between HDA-1 and DVE-1 regulated genes in the absence of stress by RNAseq?

284 genes were co-regulated by HDA-1 and DVE-1 in the absence of stress, and 374 genes were co-regulated by HDA-1 and DVE-1 under mitochondrial stress (Supplementary Fig. 2e). We determined the consensus DNA binding motifs for the two proteins. Consistent with the large overlap of genes regulated by HDA-1 and DVE-1, we found that the binding motifs were similar in the absence of mitochondrial stress and almost identical in the presence of stress (Figure 2g).

4. Also, what number of stress-activated HDA-1/DVE-1 genes are in common between the ChIPseq and RNAseq in Figure 2f and g?

Among the 283 stress-activated *hda-1*-dependent genes, 169 genes (59.7%) contain HDA-1 ChIP peaks within the ± 500 bp region of its transcription start site. In addition, among the 218 stress-activated *dve-1*-dependent genes, 175 genes (80.3%) contain DVE-1 ChIP peaks within the ± 500 bp region of its transcription start site. We included the new data and analyses in the revised manuscript (Second paragraph, Page 8, Supplementary Table 1).

5. Figure 3a: The authors use the gene *irg-1* as a readout for immune response induction. However, *irg-1* was shown to be responsive to translation inhibition and itself is not needed for protection during infection. However, the authors have uncovered a number of immune response genes whose expressions are dependent on both HDA-1 and DVE-1 (Figure 2h). Are these immune response genes induced during infection in a HDA-1/DVE-1-dependent manner (a subset of these genes would suffice)?

We thank the reviewer for the suggestion. Induction of the *irg-1* (immune response gene-1) expression has been commonly used as an indicator for the responses to *P. aeruginosa* infection. In our study, we also used *irg-1p::gfp* as a reporter to test the ability of worms to respond to *P. aeruginosa* infection. A study from the Troemel lab showed that translation inhibition alone, in the absence of infection, can activate *irg-1* expression. However, in their study, the authors also reported that the mechanism by which *C. elegans* responds to *P. aeruginosa* infection is through the detection of Exotoxin A-triggered translational inhibition

(PMID: 22520465). Therefore, they proposed that translation inhibition mimics *P. aeruginosa* infection.

We also followed the reviewer's suggestion to use qPCR to test if the immune response genes were induced in an HDA-1/DVE-1-dependent manner during *P. aeruginosa* infection. It should be noted that in Figure 2h (now Figure 2j in the revised manuscript), we analyzed the genes upregulated in an HDA-1/DVE-1-dependent manner during mitochondrial stress (*atp-2* RNAi). Among the 13 genes enriched in the "immune response" category, two of them (*pals-23* and *M04C3.2*) can also be induced by *P. aeruginosa* infection. We found that both *pals-23* and *M04C3.2* were induced in an HDA-1/DVE-1-dependent manner (Figure 3b). In addition, several other immune response genes known to be induced upon *P. aeruginosa* infection (e.g. *irg-6*, *mul-1*, *lys-2*) were also elevated in an HDA-1/DVE-1-dependent manner (Figure 3b).

6. Figure 3c: HDA-1 overexpression was found to extend host survival during infection with *P. aeruginosa* infection. Does HDA-1 overexpression also reduce pathogen colonization?

We appreciate the reviewer's suggestion. Yes, HDA-1 or DVE-1 overexpression can reduce pathogen colonization (Supplementary Fig. 4e). Because the accumulation of *P. aeruginosa* (GFP) in wild-type worms, or HDA-1 or DVE-1 overexpression worms is too low to be imaged by its fluorescence, we only performed CFU quantification.

7. How many independent transgenic lines were tested?

Two independent transgenic lines were used for the HDA-1 overexpression experiments. We observed similar results with both lines (Fig. 3f and Supplementary Fig. 4f).

8. Does DVE-1 overexpression also provide protection during infection?

Yes, DVE-1 overexpression also provides protection during infection. Compared with wild-type worms, the *P. aeruginosa* colony number was reduced in DVE-1 overexpression worms (Supplementary Fig. 4e). In addition, DVE-1 overexpression also promoted animal survival upon *P. aeruginosa* infection (Fig. 3f).

9. Figure 3d: *Pseudomonas aeruginosa*-GFP was used as a readout for colonization levels following *hda-1/dve-1* RNAi. This assay requires some form of quantification since *P. aeruginosa* colonization can be quite variable. Ideally, one would perform colony forming unit quantifications of worm lysates to accurately quantify pathogen accumulation.

We followed the reviewer's suggestion to quantify pathogen colonization. *hda-1/dve-1* RNAi increased pathogen accumulation, as revealed by CFU (colony forming unit) quantification (Fig. 3e).

10. Figure 3a and 3e: Can the authors please comment on why *irg-1::GFP* expression is fully dependent on DVE-1 during *P. aeruginosa* infection but is only partially dependent during *Rhodococcus* infection whereas HDA-1 is fully required for both?

We thank the reviewer for noticing this interesting phenomenon. Intrigued by the question raised by the reviewer, we carried out time-course experiments to thoroughly test the expression of *irg-1p::gfp* in the presence of control or *dve-1* RNAi during *P. aeruginosa* or *Rhodococcus* infection. Worms subjected to *P. aeruginosa* infection activated *irg-1p::gfp* expression faster than those under *Rhodococcus* infection. This may be due to the stronger toxicity of *P. aeruginosa*, which killed worms much faster than *Rhodococcus*. Indeed, we found that compared with *hda-1* RNAi, knockdown of *dve-1* partially suppressed *irg-1p::gfp* expression. At the current stage, we don't know why *Rhodococcus* infection is only partially dependent on DVE-1. *P. aeruginosa* is the most commonly used pathogen when studying the immune response in *C. elegans*. Its mechanism for eliciting immune response is also well studied. However, less is known about the mechanism of *Rhodococcus* infection in *C. elegans*. It will be of interest to understand how immune responses are elicited by *Rhodococcus* infection.

11. Also, is HDA-1 and DVE-1 required for survival during *Rhodococcus* infection?

Yes, HDA-1 and DVE-1 are required for survival during *Rhodococcus* infection as well (Fig. 3h).

12. Figure 4a: The authors show that *hda-1* RNAi completely suppresses the increase in lifespan of *atp-2* RNAi animals. However, *hda-1* RNAi also accelerates worm mortality by itself. Is the effect of *hda-1* RNAi on *atp-2* RNAi animal lifespan therefore specific in terms of mitochondrial stress induced longevity? In this regard, does *hda-1* RNAi suppress other long-lived animals associated with other longevity pathways (e.g. caloric restriction/*eat-2* etc.)?

We followed the reviewer's suggestion to test the effect of *hda-1* RNAi on the lifespan of *eat-2* animals. *hda-1* RNAi also shortened the lifespan of *eat-2* animals (Supplementary Fig. 5C). However, it has been reported that *eat-2* animals have decreased mitochondrial potentials (PMID: 19442682). In addition, the ZIP-2 pathway is activated in *eat-2* animals, which contributes to the improvements of mitochondrial integrity (PMID: 31215146). Therefore, it may need further analysis to understand if *hda-1* regulates lifespan through maintaining mitochondrial integrity, or through a more general mechanism.

We thank the reviewer for raising this point. We provided the *eat-2* lifespan results in the revised manuscript and added several sentences to discuss this possibility.

13. Also, HDA-1 overexpression was able to increase the survival of *C. elegans* during infection. Is HDA-1 overexpression sufficient to increase longevity as well?

We appreciate this interesting point from the reviewer. However, overexpression of HDA-1 is not sufficient to increase longevity.

14. Figure 4d: The authors claim that HDA-1 overexpression reduces polyQ aggregation, however the differences are not overtly obvious based on the images presented. Again, some quantification of the images is required. Ideally, one could compliment their visualization of

Q35::YFP aggregation by testing biochemically for changes in poly Q solubility by SDS-PAGE and Western Blotting using worm lysates. And, does DVE-1 overexpression also reduce Q35 aggregation similar to HDA-1?

We appreciate this point from the reviewer. We tried to follow the reviewer's suggestion to quantify polyQ aggregation biochemically. However, SDS-PAGE can only measure the total level of Q35. We also tried to run native PAGE, but was not able to detect polyQ aggregates. Therefore, we went back to our imaging experiments and carried out a thorough time course experiment to find the time points when the difference of polyQ aggregation was most obvious between wildtype and HDA-1 overexpression worms. HDA-1 and DVE-1 overexpression delayed the formation of polyQ aggregation. We further quantified the number of Q35 aggregates in worm muscles and provided the quantification results (Fig. 4g).

Yes, DVE-1 overexpression also delays Q35 aggregation, similar to HDA-1 (Fig. 4f, g).

15. Also, the authors only test for the effect of HDA-1 overexpression on protein aggregation (i.e. *hda-1* RNAi is only perform for the mobility assay). Is the converse true for *hda-1* RNAi?

In the revised manuscript, we showed that *hda-1* RNAi promoted the formation of polyQ aggregates (Fig. 4d, e).

16. For Figure 6b: the differences in mitochondrial morphology are not obvious in the presence or absence of HDAC1 or SATB2 during stress. Also, the authors perform all qPCRs using HEK293 cells but the morphology was performed using HeLa cells. Is there a reason for the discrepancy?

We thank the reviewer for raising this point. We initially carried out all the qPCR analysis in HEK293T cells because this cell line has a higher level of transfection efficiency, which promotes the delivery of HDAC1 or SATB2 siRNAs. We then used HeLa cells to analyze mitochondrial morphology for two reasons: 1) Compared with HEK293T, HeLa cell is an adherent cell line with larger cells which are easier to image. 2) Mitochondria tend to form more tubular structures and are organized in a complex network in HeLa cells, making it a more commonly used cell line to visualize changes in mitochondrial morphology.

We have repeated the mitochondrial morphology experiment using a concentration of antimycin and a treatment time commonly used by other labs. These conditions showed a more obvious difference in mitochondrial morphology in the presence or absence of HDAC1 or SATB2 during mitochondrial stress (Fig. 6b). We also enlarged a region of the mitochondrial network, so that the differences can be clearly evaluated.

Minor revisions

- 1. Figure 2b: red arrows are not referenced in figure legend

Thank you. We have made the correction.

- 2. Figure 6b: The authors state in the Results section that HDAC2 loss had similar effects on mitochondrial morphology but HDAC2 is not represented in the actual figure.

Thank you. We have made the correction.

- 3. Supplementary Figure 2b: the differences in DVE-1::GFP expression is not obvious. Quantifications are required.

We followed the reviewer's suggestion and quantified this image (Supplementary Fig. 2c).

- 4. The authors reference to Supplementary Figure 6 in the results section which does not exist.

Thank you. We have made the correction.

Peer Review File

Reviewers' comments, second round:

REVIEWERS' COMMENTS:

Reviewer #1 (Remarks to the Author):

Overall, I think the authors did a good job addressing my main points in the revised manuscript. I support publication of the manuscript, although I have a minor point:

In the revised manuscript, the authors have analyzed the binding motifs of HDA-1 and DEV-1. In control condition, only the 3rd most enriched motif is shared by HDA-1 and DEV-1. Interestingly, in the *atp-2* RNAi group, all top 3 motifs are shared. This is consistent with the enhanced overlapping level of the binding peaks upon *atp-2* RNAi treatment. A discussion regarding this binding shift should be included.

Reviewer #2 (Remarks to the Author):

I am satisfied with the revised copy of the manuscript by Shao, Peng et al. The authors completed all of the suggested experiments which I believe further strengthens an already well executed manuscript. I therefore support its acceptance.

Referee #1 (Remarks to the Author):

Overall, I think the authors did a good job addressing my main points in the revised manuscript. I support publication of the manuscript, although I have a minor point:

In the revised manuscript, the authors have analyzed the binding motifs of HDA-1 and DEV-1. In control condition, only the 3rd most enriched motif is shared by HDA-1 and DEV-1. Interestingly, in the *atp-2* RNAi group, all top 3 motifs are shared. This is consistent with the enhanced overlapping level of the binding peaks upon *atp-2* RNAi treatment. A discussion regarding this binding shift should be included.

We have followed the reviewer's advice to discuss the changes of the binding motifs in the revised manuscript (page 7).

Referee #2 (Remarks to the Author):

I am satisfied with the revised copy of the manuscript by Shao, Peng et al. The authors completed all of the suggested experiments which I believe further strengthens an already well executed manuscript. I therefore support its acceptance.

We thank the reviewer for the encouraging remarks on our work.